# Qualitative study of patient views on a 'telephone-first' approach in general practice in England: speaking to the GP by telephone before making face-to-face appointments

Sarah L Ball,[1] Jennifer Newbould,[1] Jennie Corbett,[1] Josephine Exley,[1] Emma Pitchforth,[1,2] Martin Roland[3]

¹Cambridge Centre for Health Services Research, RAND Europe, Westbrook Centre, Cambridge, UK
²University of Exeter Medical School, University of Exeter, Exeter, UK
³Cambridge Centre for Health Services Research, Institute of Public Health, University of Cambridge, Cambridge, UK

**Correspondence to**
Dr Sarah L Ball;
sarahb@rand.org

## ABSTRACT

**Objective**  To understand patients' views on a 'telephone-first' approach, in which all appointment requests in general practice are followed by a telephone call from the general practitioner (GP).

**Design**  Qualitative interviews with patients and carers.

**Setting**  Twelve general practices in England.

**Participants**  43 patients, including 30 women, nine aged over 75 years, four parents of young children, five carers, five patients with hearing impairment and two whose first language was not English.

**Results**  Patients expressed varied views, often strongly held, ranging from enthusiasm for to hostility towards the 'telephone-first' approach. The new system suited some patients, avoiding the need to come into the surgery but was problematic for others, for example, when it was difficult for someone working in an open plan office to take a call-back. A substantial proportion of negative comments were about the operation of the scheme itself rather than the principles behind it, for example, difficulty getting through on the phone or being unable to schedule when the GP would phone back. Some practices were able to operate the scheme in a way that met their patients' needs better than others and practices varied significantly in how they had implemented the approach.

**Conclusions**  The 'telephone-first' approach appears to work well for some patients, but others find it much less acceptable. Some of the reported problems related to how the approach had been implemented rather than the 'telephone-first' approach in principle and suggests there may be potential for some of the challenges experienced by patients to be overcome.

## INTRODUCTION

Increasing demand for general practice care is leaving practices in the UK struggling to meet patient need.[1] In response, some practices (at least 150 in England) have adopted a novel 'telephone-first' approach to managing patient requests for a consultation. In this whole system approach, all appointment requests are followed by a telephone call from the general practitioner (GP). Either the issue is resolved during this call through provision of advice, a prescription or redirection to another health professional, or the patient is invited for a face-to-face consultation, usually on the same day.

Currently, two commercial companies (Dr First and GP Access) promote this approach in the UK and provide management support to practices adopting it. The approach has been advocated by National Health Service (NHS) England, based on significant benefits reported by the companies, including improved access to primary care, improved patient satisfaction and reductions in both primary and secondary care utilisation.[2] However, an independent evaluation that we carried out found no evidence of an overall reduction in GP workload, no evidence of reduced secondary care costs and, while patients were able to be seen much more quickly, there was little overall improvement in patient satisfaction as expressed in patient surveys.[3]

### Strengths and limitations of this study

► Participants included a wide range of patients and carers from a diverse group of practices.
► Patients and carers selected for interview had recent experience of the 'telephone-first' approach.
► Participants were purposively sampled to include a wide range of views on the new approach.
► Semistructured interviews allowed participants to discuss in detail their own experiences of the 'telephone-first' approach.
► Practices agreeing to take part in the study may have been operating the 'telephone-first' approach more successfully than those that declined.

While published studies on patient satisfaction with GP telephone consultations in general report positive findings,[4–6] the 'telephone-first' approach is a much more fundamental innovation in service provision, and the National Association for Patient Participation has raised a range of concerns and opposition to the approach.[7] The patient surveys described in a previous paper[3] elicited a wide range of views about the telephone-first approach, from strongly positive to strongly negative. In this paper, we report the findings of qualitative interviews conducted with patients and carers to explore these views in greater depth.

## METHODS

### Site selection, sampling and recruitment

Qualitative interviews with patients were undertaken in 12 GP practices using the 'telephone-first' approach. Participating practices came from areas of England including the North East, North West, Midlands, East Anglia, London, the South East and the South West. Practices were selected purposively from the 20 practices participating in a patient and carer survey as part of our wider evaluation[2] to include those with a range of experiences of adopting the 'telephone-first' approach, including practices reporting positive experiences and those that had experienced or overcome problems.

In the first instance, patients who were potential participants indicated their interest in being contacted for an interview by returning a reply slip that accompanied the patient and carer survey. Purposive sampling of those who expressed an interest was carried out by the research team, to gain a range of views and to ensure people with the following characteristics were included: older people, people who work, people with disabilities, people with chronic conditions and those with English as a second language. Selected interested participants were contacted by a member of the research team by the preferred contact mode indicated in the reply slip (telephone or email) and invited to take part in an interview.

### Data collection

Semistructured interviews were conducted by four researchers (SLB, JN, JC, JE), either at the patient's home or at their GP surgery, as requested by the patient. All interviewees gave written consent to be interviewed. A common interview guide, informed by the literature, was used for each interview (see online supplementary appendix 1), although emphasis was given to allowing participants to talk from their own perspective and elements of the guide were developed iteratively as the study progressed. The main focus of the interview was on patients' and carers' views of the advantages and disadvantages of the 'telephone-first' approach, including its convenience, perceptions of quality of care and impacts on the doctor–patient relationship. Interviews were audio-recorded with the participants' permission and were transcribed verbatim. Transcripts were anonymised by removing references to identifiable names and places.

### Data analysis and reporting

Data analysis proceeded in parallel with data collection and informed the iterative development of the interview topic guide. Thematic analysis of the data was conducted based on the principles outlined by Boyatzis.[8] Transcripts were read and reread and 'codes' applied to meaningful sections of text. Coding was conducted by SLB, JC, JE, JN and EP. As analysis progressed, codes were grouped into overarching or organising themes using NVivo 10 software. Data within themes were scrutinised for confirming and disconfirming views across the range of participants. Emerging findings were shared and discussed regularly within the study team. We have followed SRQR reporting guidelines.[9]

### Patient involvement

A study steering group was established, which included four patients along with healthcare professionals. The steering group met on three occasions and provided input into the design and conduct of the study, including advice on patient materials produced during the study. Patient representatives from the steering group and those from participating practices attended a learning event at which practices shared their experiences of the 'telephone-first' approach and commented on our findings to inform their interpretation.

## RESULTS

Interviews were conducted with 43 patients and carers registered at 12 GP practices across England, all of which had been using the 'telephone-first' approach for between 18 months and five years. Respondents were aged between 28 years and 86 years and included older people, parents of young children, carers, working people and a those from a number of other 'hard to reach' groups (table 1).

**Table 1** Characteristics of interview participants

| Characteristics | Number of interview participants (%) |
| --- | --- |
| Female | 30 (69.8) |
| Aged over 75 | 9 (20.9) |
| Parent of child under 13 years | 4 (9.3) |
| Carer* | 5 (11.6) |
| Working | 11 (25.6) |
| Hearing impaired | 5 (11.6) |
| First language not English | 2 (4.7) |
| Living with a chronic condition | 24 (55.8) |
| Total | 43 |

*All five carers were interviewed in both their capacity as a carer and as a patient.

The practices at which the patients were registered varied with respect to: list size, geographical location and a range of characteristics of the catchment population, such as deprivation and ethnicity. While there were common elements to the 'telephone-first' approach used across the practices under study, there was also significant variation with respect to exactly how the approach was implemented. The characteristics of the practices, the specifics of the 'telephone-first' approach used within them and further details on the characteristics of the patients and carers interviewed are outlined in online supplementary appendix 2.

Interviews provided a rich source of data, and patients and carers were open in expressing their views (whether enthusiastic, ambivalent or hostile towards the approach). While the majority of patients, when asked to make a choice, said that they would stick with the 'telephone-first' approach rather than return to the system that their practice had run previously, responses were nonetheless extremely varied: some patients reported being highly satisfied (giving strong endorsements), while others found the approach unacceptable. In describing their experiences of the approach, patients outlined a broad range of advantages and disadvantages in relation to its impact on how they were able to access care and the nature or quality of the care received. A number of themes arose in the analysis, which we present below.

### Impact on initial contact with the practice

A clear theme was the impact of the 'telephone-first' approach on the nature of the initial contact made with the practice when booking an appointment. The perceived impact varied, with some patients describing how the initial contact was more streamlined following the changes, while others reported difficulties with or objections to the new booking process (such as long waits for calls to be answered or restricted opening times for telephone lines). For example, one patient, among a number who reacted with hostility to the introduction of the approach, described a situation in which it had taken days to get through to the practice to make an appointment:

> …. tried for two days, press five [for automatic redial] still off - and on the Thursday someone actually answered. […] Said 'what is it?' so I said what [was wrong] and I need to see the Doctor. They phoned me back then. She says well Doctor [name redacted] is not in today - phone tomorrow. Bump [phone being hung up]. So I phoned the next morning 8 o'clock. Phones off. I phoned every five min till 8.30am - it came on, 'surgery's now full', phone Monday. […] You should try the system… It's that bad you couldn't make it up. If they had someone to report it to I'd prosecute them. They're terrible. (110_1026 – male patient in his 70s, retired, multiple chronic conditions and mental health issues)

Patients attributed difficulties getting through to the practice to the way the approach had been implemented (such as a lack of reception staff to answer the telephone or shutting the phone lines early), but also highlighted that the issues caused particular problems for them as individuals, for reasons such as lacking the time to wait to get through on the telephone, or difficulty calling at the required time of day (particular issues for working people), or as a result of a personal preference for making the initial contact with the practice in person rather than by telephone.

> I just don't like it [the 'telephone-first' approach]. […] I just want a doctor's where I can go in, phone up, whatever which way I want to do it, book an appointment and go. (103_1042 – female patient in her 50s, not in employment, mental and chronic physical health problems)

### Responsiveness of the practice to patient needs

A further theme related to the perceived impact of the 'telephone-first' approach on the degree to which the practice was able to be responsive to patient needs. Patients at some practices commented positively on the prompt response of GPs following their initial call to the surgery. Guaranteed same-day call-backs (in some cases within minutes or within an agreed time slot) reassured patients who were anxious about what might be wrong with them, and the availability of timely face-to-face appointments (if required) was appreciated:

> This way I find if he [the GP] deems it serious enough for you to call in to see him, he'll see you the same day, which is brilliant. (100_1004 – female patient in her early 70s, retired, multiple chronic health issues)

> Apart from just jumping in the car and going walking into a doctor's, there's no other way you could improve that. (101_1002 – male patient in his 70s, retired, minor health issues requiring specialist input, hearing impairment)

Patients at some practices, however, described a delayed or unpredictable response, with no indication as to when the doctor would call back, or a lack of availability of appointments after speaking to the doctor. Variability between the reports of patients registered at different practices indicated that there were variations in the way the call-back system was managed, or in the capacity of practices to adequately meet demand (with respect to the availability of sufficient appointment slots for both telephone and face-to-face appointments).

In addition, interviewees described how their own personal characteristics or circumstances meant that they found unpredictability (with respect to receiving a response from the practice) particularly difficult, including a patient whose job as a support worker meant that she was unable to access her mobile phone during a shift and patients with mental health issues who reported feeling anxious or distressed while they waited for a

response from the GP. There was some acknowledgement among patients commenting on the inconvenience of having to wait for a call-back, that this had not been a particular issue for them, but indicated concern that it would be an issue in case of an urgent need.

### Implications for equitable/fair access to care

Patients were aware of and expressed strong views with respect to the implications of the 'telephone-first' approach for fairness and equity in access to care. Some patients interviewed indicated that they appreciated that the 'telephone-first' approach led to more efficient use of resources and improved access for patients with the greatest need for urgent care, and recognised that this in turn conferred benefits to them as individuals (ensuring prompt access if required):

> You get to speak to a doctor before you go in for your appointment, because I think there are a lot of times when you actually don't need to see a doctor face to face but, sometimes the advice of a doctor can put your mind at ease or just give you the information that you need to know - so then, you are not wasting your time and you are not wasting their time. (117_1066 – female patient in her 30s, single mother in part time work, infrequent user of GP)

> It's better this way because then you don't get any timewasters. […] Then you haven't got to wait. They put you first before the timewasters. (105_1043 - Female patient, mother of young child, both with chronic health issues)

Patients differed in their perceptions of the intended function of the approach with respect to redirecting patient demand. While some patients perceived the 'telephone-first' approach to be a fair system for meeting patient need, others saw it as a barrier, intended to keep as many patients as possible away from face-to-face appointments with busy doctors, describing feeling the need to '*fight*' or '*protest*' to justify their requests to see a doctor face to face:

> It certainly feels like a gate-keeping service […] like being kept as much at arm's length as possible. (114_1058 – female patient in her early 70s, retired, chronic health issues)

Some expressed concern on behalf of vulnerable patients, such as the elderly or those with mental health issues, who may lack the confidence or communication skills to push for an appointment when required.

### Ease and convenience of access to care

A strong theme in the analysis centred on how the introduction of the 'telephone-first' approach had affected the ease and convenience with which patients were able to access care. For some, the new mode of access had resulted in increased convenience, while for others the opposite had been the case. Commonly, the patients interviewed reported that they found that the approach enabled more convenient access to advice and care than the system previously in place, with benefits including: being able to get on with daily activities while waiting for a response from the practice, rather than having to wait for long periods in the surgery (facilitated by the availability of mobile telephone contact); reduced need to travel to the surgery unnecessarily (a particular benefit for those for whom travelling to the surgery was difficult, such as a mother with disabled children, a carer whose husband was disabled with chronic conditions and mobility issues and those dependent on limited public transport services); and access to medication without the need for a face-to-face appointment:

> I like the fact that on a day like today, it is chucking it down, it's miserable, it's cold, if my mum had had to come to the doctor instead of a phone call on any day where the weather was like this, it would have caused her a lot of pain. (102_1031 – female patient in her 40s, works part time, ongoing mental and physical health issues)

In particular, patients able to accommodate time constraints of the approach (eg, being at home during the day, retired or working flexibly) highlighted how the 'telephone-first' approach fitted conveniently with their daily schedules. Others, however, found the approach inconvenient in one or more respects, including: not being able to book appointments in advance; receiving a call from the GP at inconvenient times (when shopping, on public transport or at work); or having to stay at home to wait for a call, particularly if it related to a personal issue that it was difficult to discuss in public:

> You can't sit glued to your phone all day waiting for a call, even if you've got a mobile phone, you might be in the shower, or you might be in a shop or on the other phone or something. So it doesn't work… and how people who are working, expect to get an appointment I don't know really. (110_1007 – female patient, early retirement due to ill health)

Patients at several practices, described how such issues had been addressed by their practice by ensuring flexibility in the approach, such as by accommodating patient requests for a call-back at a particular time or offering limited advanced bookings for those unable to attend on the same day.

Similarly, patients with particular difficulties that had an impact on how they were able to interact with the practice using the 'telephone-first' approach, described how such difficulties were overcome by minor adjustments and a flexible approach, for example, special arrangements for patients whose first language was not English or those with a hearing impairment.

### Differences in the nature of GP consultations: efficiency, communication and social contact

Patients highlighted differences in the nature of GP consultations as a result of the new approach, identifying

differences between telephone and face-to-face consultations and the impact of initial telephone contact on subsequent face-to-face appointments. Patients described both advantages and disadvantages of initially consulting by telephone rather than face to face, reflecting individual differences with respect to confidence and efficacy of communication by telephone and the value placed on face-to-face contact with a GP.

While some patients described feeling very comfortable communicating by telephone, including some patients with mental health issues who preferred telephone consultations because they felt more relaxed, others reported difficulties describing symptoms or understanding and recalling the GP's advice. Those patients reporting that they felt anxious when communicating on the telephone included older people, those with mental health issues, hearing impairment and one for whom English was not his first language. Others reported concerns on behalf of other patients:

> I've got a friend, an old lady who's 88, going on 89, I think, and she absolutely hates [it]. She says 'I can't talk on the phone, I just don't know what to say, I just go to pieces.' And somebody like her, it's just totally awful you know, it's not satisfactory at all. (110_1007 – female patient in her 60s, early retirement due to ill health)

Some interviewees commented that they found the approach to be impersonal, resulting in consultations that were rushed and to the point, an issue that was highlighted by patients with mental health concerns and chronic conditions in particular. This was in part attributed to a lack of relational continuity of care (see below) but also due to the nature of the telephone consultation itself and the absence of the social cues present in face-to-face interaction. A patient with mental health issues described the negative impact of a lack of face-to-face contact on the nature of the consultation:

> I just cannot cope with not seeing someone's face […] I just want to speak in a room with the door closed face-to-face with someone so that I can be honest about how I am feeling and what's been happening lately. So I don't really say much over the phone […] whereas if it was face-to-face I would explain more (110_1095 – female patient in her 60s, part-time work, ongoing mental health issues)

Changes in the nature of face-to-face appointments following the introduction of the 'telephone-first' approach were also noted, including improvements such as reduced waiting time in surgery and a calmer more relaxed atmosphere, with patients experiencing less time pressure during appointments. Some patients suggested that the approach led to GPs being better prepared and the appointment being more streamlined as a result. A few among those who did not observe any difference in the nature of face-to-face appointments, however, commented that having to repeat details given over the telephone in the face-to-face appointment was an annoyance and appeared inefficient.

## Effects on continuity of care

Given claims made by commercial providers that the 'telephone-first' approach can improve continuity of care for patients, interviewees were asked specifically about changes in the ease with which they were able to see a preferred GP. Again, there was variation between participants in their responses. Some patients reported finding it easier to see or speak to their preferred GP than with the previous system, as a result of the way in which calls were allocated within the practice, with patients being able to specify which GP they would like to call them back. If this was not possible, they could request a face-to-face appointment with the preferred GP during the telephone call. Others, however, reported the opposite and found it harder to see their GP of choice, observing a trade-off between being seen or spoken to quickly and seeing their preferred GP. These observations highlighted differences between practices introducing the 'telephone-first' approach with respect to their capacity to preserve or enhance continuity of care.

In addition, the degree to which the patients and carers interviewed were concerned about the impact of the introduction of the 'telephone-first' approach in this regard, varied between patients, according to the value they placed on their relationship with a particular GP. Concern was expressed about whether an unfamiliar GP could effectively assess an issue over the telephone and some patients worried about the lack of opportunity to develop or sustain a relationship with a GP (a particular concern among patients with chronic conditions and those with ongoing mental health issues):

> [an unknown] GP rang me back and I wasn't sure whether he knew anything about me. I'm quite sure he'd looked at my records very briefly but I was concerned because it's quite complicated and my preferred GP knows from day one and has worked with me and referred me and supported me, so I didn't know how much this person knew and I just was a little bit unsure and a little bit anxious about whether or not he knew enough about me (110_1095 – female in her 60s, part-time work, ongoing mental health issues)

## Implications for patient safety

Patients speculated on the implications of the 'telephone-first' approach with respect to patient safety. Views on the impact (or potential impact) of the approach in this regard varied considerably between patients and across practices. While some patients felt vulnerable because of difficulties getting through to the practice by telephone or the fear that diagnoses might be missed in telephone consultations, others thought the approach was safer for patients, in part, because of the considerable reduction in waiting times for appointments:

Well I think you get to talk to your doctor when you need to talk to him or her, rather than having a long wait and perhaps getting progressively worse. Certainly if it's an acute condition, it can make a difference, can't it? (100_1004 - Female patient in her early 70s, retired, multiple chronic health issues)

So I phoned up and it was early in the morning and I mentioned to the receptionist what the problem was, and so within minutes another doctor phoned back and he said you, had better come down. (117_1029 - Female patient in her 60s with chronic health issues, not in employment, caring responsibilities)

Concerns were expressed among patients who were currently confident in their own communication skills that being less articulate or lacking the confidence to push for a face-to-face appointment when required may put some patients at risk of not receiving treatment they needed.

### Concerns regarding confidentiality
Concerns regarding confidentiality associated with the 'telephone-first' approach marked a common theme in the analysis, as the system generally required the receptionist to ask the patient for brief details of their problem during the initial call to the practice:

[Y]ou know that whatever you say to a doctor is going to stay with the doctor, with the receptionist, you are never quite sure if it's going to stay there (117_1066 - Single mother in her 30s, part-time work, infrequent user of GP)

Strong feelings were expressed on this subject with, for example, one patient describing the approach as '*absolutely disgusting*' (103_1042). Concern was even expressed by patients who acknowledged the benefit of providing the information in order for calls to be prioritised. Patients also reported concerns about confidentiality associated with the telephone consultation itself, especially if they had to receive the call-back from the GP at a time and/or in a location where their conversation could be overheard, whether at home with family members present, in a work setting or on public transport.

### The importance of understanding the purpose of the approach and how it works
Patients described their understanding of the rationale for the introduction of the 'telephone-first' approach, how it was supposed to work in practice and how this had influenced their response to it. The degree to which patients reported that they had been consulted (or at least informed) ahead of the introduction of the new approach varied considerably. Some patients highlighted their lack of awareness about how the approach would work in practice at the outset and expressed irritation with a lack of consultation around its introduction, which had resulted in confusion, anxiety and misconceptions regarding the purpose behind the introduction of the

approach. Others, however, commented that their initial misgivings had not, by and large, been realised.

Patients indicating their awareness that the approach involved the prioritisation of calls according to need acknowledged the necessity of waiting for a call-back accordingly:

I mean sometimes if he's [the GP] really busy, you don't hear from him for a couple of hours but then he's obviously got patients there that are a priority. They know how to prioritize them which is good. (102_1014 – female patient, in her late 70s, retired, multiple chronic conditions)

### Assessing the overall acceptability of the approach
The advantages and disadvantages of the 'telephone-first' approach reported by patients varied between individuals and reflected both the way in which the 'telephone-first' approach had been implemented and the patients' own individual characteristics and resources. In assessing the overall acceptability of the approach, patients made reference to both these types of characteristic, and there was apparent interplay between them. Patients explained how specific issues or disadvantages resulting from how the approach had been implemented were particularly problematic for them as an individual, as a result of personal characteristics or preferences, or the structure of their daily life. For example, having a long or unpredictable wait for a call-back from a GP was an issue for patients unable to access a mobile phone or find a quiet, private place to take a call at work but a lesser concern for those who were retired or who were able to work flexibly. Examples of the kinds of factors considered by the patients interviewed as they assessed the overall acceptability of the 'telephone-first' approach are presented in tables 2 and 3.

The value attributed to particular advantages and disadvantages varied significantly, even between patients from the same practice or with similar characteristics. A disadvantage that represented a mild annoyance for one patient could render the approach completely unacceptable for another. For example, one patient experiencing mental health issues described the effect of having to wait a long time for a call-back from a GP while in a distressed state and how this had influenced her decision to leave the practice:

I was really low and so I think I had to wait a few hours [for a call from the GP] and all that time I was in tears and it still took a couple of hours for the doctor. I thought, 'Well, now I can't be bloody bothered'. (103_1042 – female patient in her 50s, not in employment, mental and chronic physical health problems)

In addition, interviewees also acknowledged that some issues such as difficulty with getting through to the practice initially, or long waits for a response from the practice were (or had the potential to be) of greater concern in some instances than in others, dependent on

**Table 2**  Practice/system characteristics that influenced patients' assessment of the acceptability of the 'telephone-first' approach

| Patient characteristic or resource | Factors influencing patients' assessment |
|---|---|
| Communication skills | The degree to which they feel able to adequately communicate over the telephone. |
| Confidence | The degree to which they feel confident to request the outcome they want. |
| Flexibility of daily schedule | The degree to which they are able to accommodate time constraints of the approach for example, being at home during the day/retired/ working flexibly. |
| Access to mobile telephone | Whether they are easily accessible on a mobile telephone. |
| Value placed on face-to-face contact with GP | The value they place on face-to-face contact compared with ease and speed of access to care. |
| Nature of relationship with GP or surgery | The value they place on a long-standing, trusting relationship with a GP. |
| Nature of the reason for contacting the surgery | Perceptions regarding the urgency of the issue for which they are seeking care. |

the perceived urgency of the issue for which they were seeking care.

In describing their assessment of the overall acceptability of the approach, some patients recounted long lists of annoyances (difficulty getting through on the telephone, confidentiality concerns when talking to receptionists, not being able to book in advance and not liking waiting for the call-back) but still concluded that they preferred the new approach, because they could speak to a doctor within hours and see them the same day if they needed to (an outcome on which they placed particular value).

## DISCUSSION

The study showed that, consistent with our published quantitative analysis of the patient and carer survey in our evaluation,[2] patients expressed a wide range of views, often strongly held, on the 'telephone-first' approach.

Qualitative interviews allowed us to understand these views in greater depth and to explore some of the reasons behind the different views expressed. The new system clearly suited some patients, (for example, by allowing them to avoid coming into the surgery) but was problematic for others (for example, when it was difficult for someone working in an open plan office to take a call-back from the GP). Variation was evident within as well as between the different patient groups we recruited from and appeared to be influenced by the interplay of individual and practice level characteristics. Notably, a substantial proportion of negative comments were about the operation of the scheme itself rather than the principles behind it, for example, difficulty getting through on the telephone or being unable to schedule when the GP would call back. Some practices were able to operate the scheme in a way that met their patients' needs better than others and practices appeared to vary significantly

**Table 3**  Individual characteristics and resources that influenced patients' assessment of the acceptability of the 'telephone-first' approach

| System/practice characteristic | Factors influencing patients' assessment |
|---|---|
| Capacity of the system to meet demand | Whether telephone calls to the practice are answered promptly. |
|  | Whether there are sufficient appointment slots available for both telephone and face-to-face appointments. |
| Flexibility of the approach | Whether advanced booking is available. |
|  | The degree to which there is flexibility in the timing offered for the GP to call back or ability to book the time of the call-back |
|  | Whether patients are required to describe their problem to the receptionist. |
|  | Whether adjustments have been made for patients who found difficulty with the approach. |
| Capacity to preserve or enhance continuity of care | Whether a choice of GP offered for telephone consultation and subsequent face-to-face appointment. |
| Extent of patient education/knowledge | Whether patients were consulted prior to introducing the approach. |
|  | Whether clear and updated instructions had been provided on how the system works. |

GP, general practitioner.

in how they had implemented the approach, according to patients' accounts.

The NHS in England has prioritised improving access to care for several years and the 'telephone-first' approach is one attempt to address access problems, while at the same time trying to avoid an increase in practice workload. The finding in this study that the approach has been positively received by many of the patients interviewed is supportive of previous research indicating that there is considerable potential for using telephone consultations in general practice.[5 10] Our findings also chime to a degree with previous findings suggesting that access is not the main driver of patients' satisfaction with their GP practices, with interpersonal aspects of care and helpfulness of receptionists being more important[11 12] (although our findings suggest that the value placed on the different aspects of care may vary considerably between patients, according to their individual needs and preferences).

The study highlights the need for clinicians and policymakers to take the needs of patients with varying care-seeking and interaction approaches into account when making major changes to the organisation of general practice care. This change, while designed to improve access to care and reduce the workload burden on practices, clearly did not meet the needs of all patients and provoked outright hostility in some, particularly among those who struggled to access care at all as a result of issues with how the scheme had been implemented. Practices considering making this change should reflect on how they can make the scheme flexible for patients' needs, how they can make it easy for patients to get through on the telephone and how they can use the approach to enhance both access and continuity of care and recognise the need for continued development and adaptation of the approach.

A strength of the study is that the interviews included a wide range of patients and carers from a diverse group of practices, purposively sampled to capture a variety of views on the new approach. However, a limitation is the likelihood that practices operating the 'telephone-first' approach successfully were more likely to participate in the patient survey that provided patients who volunteered to be interviewed. We do not know how the views of patients participating in the study may compare with other patients, including those in practices that have not implemented the 'telephone-first' approach.

A question that could be addressed by future research is how to develop systems that are flexible enough to meet the needs of all their patients. While a rigid 'telephone-first' approach for all consultations does not do this, we observed practices that were modifying this approach (by, for example, allowing for some advanced booking of appointments) often on an ongoing basis, to meet the needs of patients as closely as they could. Successful approaches are likely to be different in different practices and more work could be done to identify what works best in different circumstances and to share learning.

## CONCLUSIONS

The 'telephone-first' approach appears to work well for some patients, but others find it much less acceptable. Some of the reported problems related to how the approach had been implemented rather than the 'telephone-first' approach in principle and suggests there may be potential for some of the challenges to be overcome. A range of factors were identified that should be considered by practices planning the approach in order to maximise its acceptability and best meet the needs of patients.

**Acknowledgements** We would like to thank the patients and carers interviewed for this study and those who completed the questionnaires that provided the basis for participant recruitment. We are grateful to the general practitioners (GPs) and staff at practices taking part in the study for their support with its conduct. We would also like to thank the GPs, practice manager and patients on the study steering group who gave guidance on the design and conduct of the study and all those who attended and contributed to study learning events.

**Contributors** All authors contributed to the conception or design of the work and the interpretation of the findings. MR is Principal Investigator for the study and had oversight, JN was project lead and SLB was project manager. SLB, JN, JC and JE were involved in data collection. SLB, JN, JC, JE and EP conducted data analysis. All authors were involved in drafting and commenting on the paper and have approved the final version.

**Funding** The study was funded by the National Institute for Health Research (HS&DR Project 13/59/40). Part of the funding was used to pay for data to be extracted from practice records by one of the commercial companies providing management support for the 'telephone-first' approach (GP Access). GP Access had no input into the analysis or interpretation of the data. The study was sponsored by Cambridgeshire and Peterborough Clinical Commissioning Group (CCG), who gave initial approval for the project.

**Disclaimer** This article presents independent research funded by the National Institute for Health Research (NIHR). The views expressed are those of the authors and not necessarily those of the NHS, the NIHR or the Department of Health.

**Competing interests** None declared.

**Patient consent for publication** Not required.

**Ethics approval** The study was approved by the West of Scotland NHS Research Ethics Service (7 May 2015, REC reference 16/WS/0088).

**Provenance and peer review** Not commissioned; externally peer reviewed.

**Data sharing statement** No additional data are available.

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
