## [Reviewer comments · BMJ Open]

ARTICLE DETAILS

TITLE (PROVISIONAL)	A qualitative study of patient views on a 'telephone-first' approach in general practice in England: speaking to the GP by telephone before making face-to-face appointments
AUTHORS	Ball, Sarah L.; Newbould, Jennifer; Corbett, Jennie; Exley, Josephine; Pitchforth, Emma; Roland, Martin

VERSION 1 – REVIEW

REVIEWER	Helen Atherton University of Warwick, UK
REVIEW RETURNED	11-Sep-2018

GENERAL COMMENTS	Thank you for the opportunity to review this interesting manuscript in a very important area - it is timely to see this research at a time when more general practices are adopting this method of giving patients access. This is a well written manuscript with important conclusions. There are, however, some modifications that need to be made to the results section to better reflect the qualitative methodology used. My comments on each section are outlined below. Abstract, introduction - no comments Methods - page 4, line 24 - you state you seek to identify characteristics of both practices and patients that have influenced acceptability. please can you describe how you did this in the context of a qualitative study? Did you apply a theoretical framework? Did the analysis focus on acceptability? At present it reads as though you intended to draw conclusions on the basis of numbers of patients which would not be possible with this dataset. Results Currently the study results are presented in a very quantitative style, using terminology such as 'several patients,' 'more than a quarter' 'a number of patients' etc. In qualitative research the aim is not to count views, or perceptions, and the write up doesn't fit with thematic analysis, which was the methodology used. Something is lost in the very categorical way you present the data. Ideally you would present the themes as derived from the data, not referring to the number of patients as this is not relevant in a qualitative study. I haven't put any detailed line by line comments here, as I feel it would easier if you just review this section and present it according to the results of the thematic analysis, starting by telling the reader what themes you found and how the data supports these (or where there are disconfirming cases) and not counting or referring to categories. These terms are much more
--

	associated with the coding stage of thematic analysis, and by the point of the write up you should have refined your findings. Tables 2 &3 I felt uncomfortable with the creation and use of these tables from the data you present. As you have not used a theoretical framework on which to build your theories about what works and what does not, using the data as presented, to produce lists is not very useful. You could make suggestions based on the analysis of your sample, but this would not be definitive and so probably isn't worth the word count, which you could use to elaborate on the findings. Discussion/conclusions The manuscript draws conclusions that are not currently supported by the results. Page 12, line 52 talks about your study showing considerable potential for using telephone consults and access not being main driver of satisfaction - but I didn't get that clearly in your results, I didn't see a theme that addressed access in quite this way. The conclusions are strong and this is a very important paper that just needs some tweaks.
--	--

REVIEWER	Claire Jackson UQ Australia
REVIEW RETURNED	07-Oct-2018

GENERAL COMMENTS	Good follow up paper, expanding knowledge base of previous research in an important area of primary care reform of interest internationally. Thorough description of findings. Sound method and care taken with sampling of both practices and patients. Good justification and useful discussion and application No major concerns or requirements
---

VERSION 1 – AUTHOR RESPONSE

Comment source	Comment	Our response
Reviewer 1	Methods Page 4, line 24 - you state you seek to identify characteristics of both practices and patients that have influenced acceptability. please can you describe how you did this in the context of a qualitative study? Did you apply a theoretical framework? Did the analysis focus on acceptability? At present it reads as though you intended to draw conclusions on the basis of numbers of patients which would not be possible with this dataset.	The method section has been amended. The statement on page 4, line 24 on 'seeking to identify characteristics of both practices and patients that have influenced acceptability' has been deleted. We did not apply a theoretical framework. 'Assessing the overall acceptability of the approach', was in fact a theme that came out of the analysis.

		Patients described how their individual needs and characteristics, practice characteristics, and specific features of the 'telephone first' approach as implemented influenced their overall assessment of acceptability of the approach. This is now written up as a theme in the results section.
Reviewer 1	Results Currently the study results are presented in a very quantitative style, using terminology such as 'several patients,' 'more than a quarter' 'a number of patients' etc. In qualitative research the aim is not to count views, or perceptions, and the write up doesn't fit with thematic analysis, which was the methodology used. Something is lost in the very categorical way you present the data. Ideally you would present the themes as derived from the data, not referring to the number of patients as this is not relevant in a qualitative study. Review this section and present it according to the results of the thematic analysis, starting by telling the reader what themes you found and how the data supports these (or where there are disconfirming cases) and not counting or referring to categories	The results section has been substantially rewritten to address reviewer 1's concerns regarding the quantitative style of presentation used. Care has been taken to avoid quantification where possible, - with two exceptions:  - the use of the term 'few' to denote a minority view - the inclusion of quantitative descriptors in paragraph 3 of the results – providing an overview of interview characteristics and the variability of the sample with respect to their overall reaction to the telephone-first approach. Themes as derived from the data are now presented as suggested, without reference to categories, with description of how the data supports each theme.
Reviewer 1	Tables 2 &3 I felt uncomfortable with the creation and use of these tables from the data you present. As you have not used a theoretical framework on which to build your theories about what works and what does not, using the data as presented, to produce lists is not very useful. You could	In response to the reviewer's comment, we have refocused tables 2 and 3 so that they present the key factors that the patients interviewed considered when determining overall acceptability of the telephone

	make suggestions based on the analysis of your sample, but this would not be definitive and so probably isn't worth the word count, which you could use to elaborate on the findings.	first approach. This is based directly on the data presented, (rather than going beyond this to theorise about what works and what does not). We would argue that bringing together these considerations in tabular format is helpful to the reader, particularly given the need to orient findings to a policy audience, and provides a sense of the issues practices introducing (or thinking about introducing) the approach should consider in order to best meet patient needs.
Reviewer 1	Discussion The manuscript draws conclusions that are not currently supported by the results. Page 12, line 52 talks about your study showing considerable potential for using telephone consults and access not being main driver of satisfaction - but I didn't get that clearly in your results, I didn't see a theme that addressed access in quite this way.	The conclusions have been refined in line with Reviewer 1's observation. Potential for using telephone consultations is supported by the finding in this study that patients report many advantages of the telephone-first approach (such as increased convenience, and prompt access to care). Findings with respect to the importance of interpersonal contact for some patients lends partial support to the studies that suggest access is not the main driver of satisfaction, although our finding would suggest that the value of these factors varies between patients. The conclusions have been amended to address this.

VERSION 2 – REVIEW

REVIEWER	Helen Atherton University of Warwick, UK
REVIEW RETURNED	12-Nov-2018
GENERAL COMMENTS	Thank you for your response to my comments and for the changes made. I've recommended this manuscript for publication.